# Integrated single-cell multiomic profiling of caudate nucleus suggests key mechanisms in alcohol use disorder

Nicholas C. Green [1], Hongyu Gao[1,2], Xiaona Chu[1], Qiuyue Yuan[3], Patrick McGuire[1], Dongbing Lai [1,2], Guanglong Jiang [1], Xiaoling Xuei [1], Jill L. Reiter[1], Julia Stevens[4], Greg T. Sutherland [4], Alison M. Goate [5,6], Zhiping P. Pang [7,8], Paul A. Slesinger [5], Ronald P. Hart [7,9], Jay A. Tischfield[7,10], Arpana Agrawal [11], Yue Wang [1], Zhana Duren [1,2], Howard J. Edenberg [1,12] ✉ & Yunlong Liu [1,2] ✉

Alcohol use disorder (AUD) induces complex transcriptional and regulatory changes across multiple brain regions including the caudate nucleus, which remains understudied. Using paired single-nucleus RNA-seq and ATAC-seq on caudate samples from 143 human postmortem brains, including 74 with AUD, we identified 17 distinct cell types. A significant portion of the alcohol-related differences in gene expression were accompanied by a corresponding difference in chromatin accessibility within the gene. We observed transcriptional differences in medium spiny neurons that impact RNA metabolism and immune response pathways. A small cluster of D1/D2 hybrid neurons showed AUD-induced differences distinct from the D1 and D2 types, suggesting a unique role in AUD. Those with AUD had a higher proportion of microglia in an inflammatory state; astrocytes entered a reactive state partially regulated by *JUND*. Oligodendrocyte dysregulation was driven in part by *OLIG2* activity and increased TGF-β1 signaling from microglia and astrocytes. We also observed increased microglia-astrocyte communication via the IL-1β pathway. These findings provide valuable insights into the genetic and cellular mechanisms in the caudate related to AUD. They also demonstrate the broader utility of large-scale multiomic studies in uncovering complex gene regulation across diverse cell types, which has implications beyond the substance use field.

Excessive alcohol use creates many serious physical, emotional, and social problems and is responsible for about 3 million deaths worldwide each year[1]. Alcohol use disorder (AUD) is a serious and common psychiatric disorder that is characterized by excessive alcohol consumption and consequent psychological and interpersonal problems stemming from preoccupation with and a loss of control over drinking[2]. The risk of developing AUD depends on both genetic and environmental factors; between 50% and 60% of the difference in vulnerability to AUD is inherited[3]. While recent large-scale genome-wide association studies (GWAS) have identified hundreds of loci associated with alcohol consumption[4,5] and AUD[6–8], it is not yet clear which genes within the loci are responsible, nor how they contribute to AUD.

In addition to inherited differences, the extended, excessive use of alcohol characteristic of AUD is likely to alter gene expression and chromatin conformation. While microarray and bulk RNA sequencing

(RNA-seq) studies have identified differentially expressed genes (DEGs) in human cell lines[9–12] and in several brain regions in rat[13–20] and human post-mortem tissue[10,21–23], bulk studies obscure the unique transcriptional signatures of individual cell types within the brain. Single-cell/single-nucleus RNA sequencing (snRNA-seq) has enabled measurement of the distribution and characterization of different cell types in a tissue sample and of gene expression within each of these individual cells. An early snRNA-seq study of nuclei from the prefrontal cortex of seven individuals identified seven major cortical cell types and found differences in expression associated with AUD within six cell types[24].

The caudate nucleus forms part of the dorsal striatum and, more broadly, the basal ganglia, a key component of the executive control loop that is recruited in the onset and maintenance of AUD[25]. The caudate has been implicated in cue-elicited activation, dopamine increase, and subjective reports of craving[26,27]. In animal models, chronic ethanol exposure alters neural circuits in the basal ganglia[28,29]. A recent study reported differences in gene expression in the dorsal striatum of alcohol-preferring rats[30]. A transcriptome-wide association study found that among 13 human brain tissues, the caudate was the region with the most genes whose predicted expression was associated with problematic alcohol use (PAU), a trait that combines AUD with problematic alcohol drinking[6].

The caudate harbors multiple cell types[31], and cell-type-specific characterization of the AUD-associated transcriptome in the human caudate is lacking. To meet this gap in knowledge, we sought to provide a comprehensive view of AUD-related differences in gene expression and chromatin accessibility in specific cell types within the human caudate nucleus and infer mechanisms underlying these changes. Single-nucleus multiome experiments (sn-multiome), assaying both chromatin accessibility and gene expression within the same cell, provide remarkable opportunities to draw causal inferences regarding mechanisms underlying AUD-associated gene expression. We performed a high-throughput snRNA-seq experiment on human post-mortem samples from the caudate nucleus of 143 donors, 74 with and 69 without AUD, obtaining transcriptomic data from over 1.1 million cells. To compare the transcriptome with the open chromatin status within the same cells, we also performed an sn-multiome (snRNA-seq + assay for transposase-accessible chromatin with sequencing (ATAC-seq[32])) analysis from these same caudate samples, allowing us to both identify uncommon cell types and unique cell states, and measure small differences in both gene expression and chromatin accessibility in the same nuclei. We profiled the biological pathways underlying these differences, and AUD-associated differences in transcription factor activity and cell-cell communication in major glial cell types (Supplementary Fig. 1). This study provides a comprehensive profile of AUD-related differences in the caudate nucleus, identifies potential mechanisms contributing to AUD and several directions for further exploration, and highlights the broader utility of large-scale multiomic studies for identifying regulatory mechanisms, which can be applied to other neurological and psychiatric conditions.

## Results

### Clustering reveals 17 well-established and well-differentiated striatal cell types

Samples from the caudate nucleus of post-mortem brains from the New South Wales Brain Tissue Resource Centre at the University of Sydney were sequenced in the sn-multiome assay, in which transcription levels and chromatin accessibility were measured in the same nuclei; most were also sequenced using the 10X HT snRNA-seq assay. After demultiplexing and data processing, samples with <200 cell barcodes were removed, leaving 163 samples, 82 with and 81 without AUD; 128 male and 34 female. Low quality nuclei were filtered out from further analyses based on number of genes, number of molecules, and percentage of mitochondrial DNA (see "Initial quality control" in "Methods"), leaving gene expression levels for 1,307,323 nuclei and chromatin accessibility (ATAC-seq) for 267,100 of these nuclei (Demographics are in Supplementary Data 1 and 2, and detailed experimental procedures are in "Methods"). Graph-based clustering of the snRNA-seq data of the 163 samples identified 17 distinct cell clusters (Fig. 1a, Supplementary Fig. 2a). There was no significant difference in relative abundance of cell types between samples from individuals with and without AUD (Supplementary Fig. 2b; see "Methods").

Three subtypes of medium spiny neurons (MSNs, the GABAergic projection neurons of the striatum) were identified: D1-type MSNs, D2-type MSNs, and a third subtype marked by both *DRD1* and *DRD2* expression (D1/D2 neurons). This cluster showed strong expression of *RXFP1*, a known marker of D1/D2 hybrid neurons[33]. Interestingly, D1/D2 cells had a similar *DRD1* expression level (average normalized expression = 83.8) to the D1 MSNs (86.2), but *DRD2* expression was far lower in D1/D2 cells (41.9) compared to D2 MSNs (85.6). Similarly, the D1/D2 cluster had a higher total number of ATAC-seq reads surrounding the *DRD1* transcription start site (TSS) (3079 within a 5 kb region) than the *DRD2* TSS (959).

Four small populations of GABAergic interneurons were identified, including parvalbumin-expressing fast-spiking (FS), neuropeptide Y/somatostatin/nitric oxide synthase-expressing low-threshold-spiking (LTS), calretinin-expressing (CR), and cholecystokinin-expressing (CCK). A small cluster of cholinergic neurons was also identified (Ach). In addition to neurons, several glial cell populations were observed, including oligodendrocytes (the most prevalent cell type; 28.2% of the nuclei), oligodendrocyte progenitor cells (OPCs), astrocytes, ependymal cells, and microglia. Other cell types included non-microglial macrophages, endothelial cells, and vascular smooth muscle cells. Unexpectedly, glutamatergic neurons were found, although the caudate is known not to contain cell bodies of excitatory neurons. Their presence could reflect inadvertent inclusion of another brain region at the time of dissection; therefore, twenty samples that contained >10% of glutamatergic neurons were removed from all subsequent analyses (Supplementary Data 1 and 2), leaving 143 samples (74 with AUD, 69 without; 115 male, 28 female) with gene expression data for 1,121,762 nuclei, 250,537 of which also had ATAC data. Our cell type annotations correspond well to the cell groups from the recent Mammalian Basal Ganglia Consensus Cell Type Atlas released by Allen Institute[34] (Supplementary Fig. 3).

### Abundance of inflammatory microglia is positively associated with AUD and age

Using data from the 143 samples, graph-based subclustering was performed for microglia, astrocytes, and D1 and D2 MSNs, the clusters that are known to have functional subtypes in the central nervous system[35–37], and had a sufficient number of cells to analyze. There were four subclusters of microglia that roughly correspond to different activation states (Fig. 1b, Supplementary Data 3 and 4). Subcluster 1 (Resting Microglia) uniquely expressed genes specific to quiescent microglia, such as *P2RY12* and *CX3CR1*, and was enriched for pathways relating to microglia migration. Subclusters 2 and 3 were both enriched for immune response-related genes, with subcluster 2 (Inflammatory Microglia) highly expressing genes involved in inflammation, such as *TLR2*. Subcluster 3 (CD83+ Microglia) was enriched for genes governing microglia activation, such as *CD83*[38]. Subcluster 4 (Phagocytosing Microglia), annotated by a recent cell type atlas as border-associated macrophages[34] was marked by high expression of genes involved in endocytosis and phagocytosis. In individuals with at least 50 microglia cells, there was a significantly higher proportion of Inflammatory Microglia (subcluster 2) in individuals with AUD: 31%, versus 23% in those without AUD (false discovery rate (FDR) = 0.027).

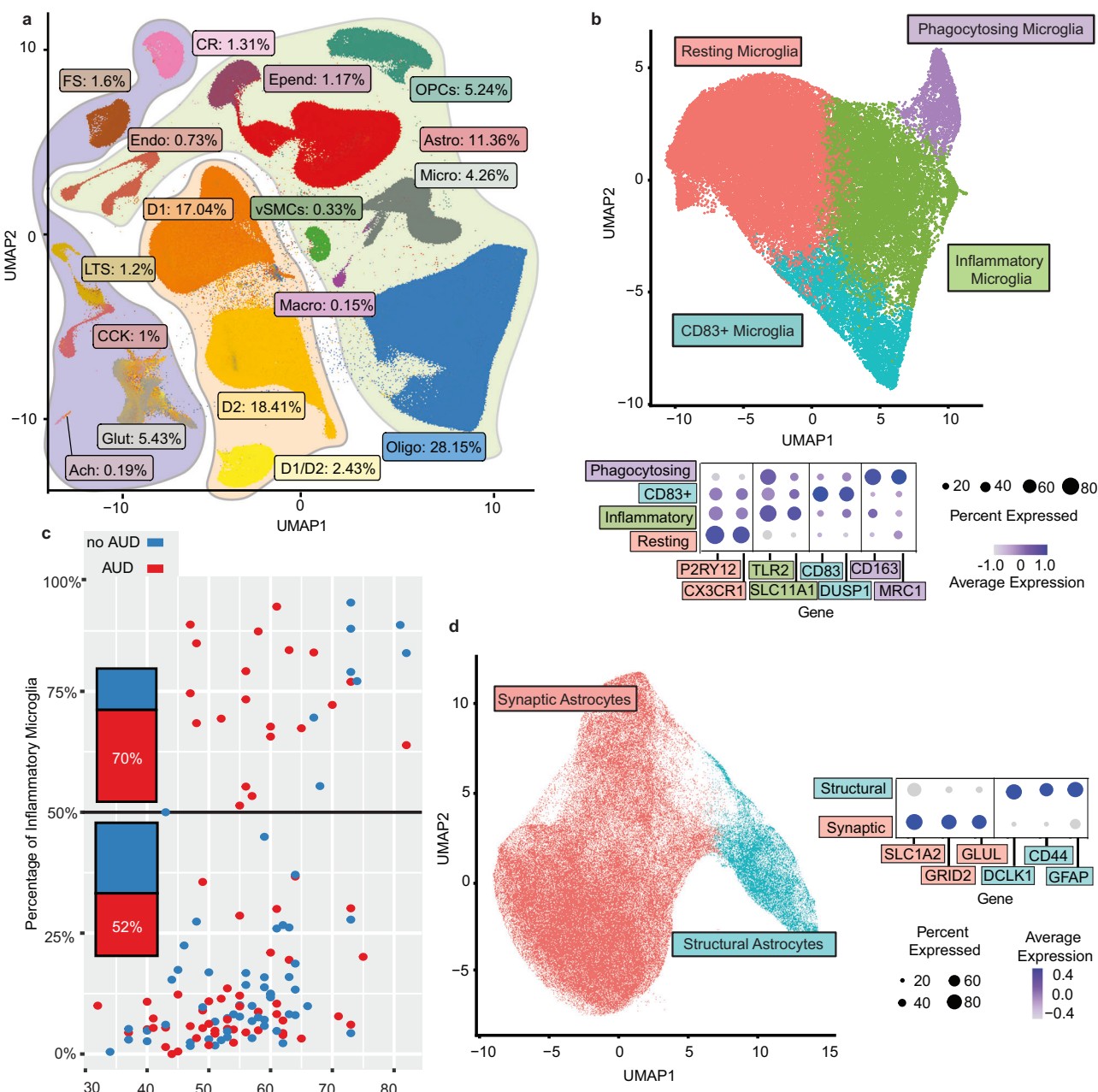

**Fig. 1 | Cell type landscape of the caudate nucleus in alcohol use disorder.**
**a** UMAP plot of the 1,307,323 nuclei profiled in the snRNA-seq and sn-Multiome assays; visualization shown is based on the snRNA-seq profile. Nuclei are labeled by cell type and cell type proportion among all snRNA-seq cells. Cell types: cholinergic neurons (Ach), astrocytes (Astro), cholecystokinin-expressing interneurons (CCK), calretinin-expressing interneurons (CR), D1-type medium spiny neurons (D1), D2-type medium spiny neuron (D2), medium spiny neurons expressing both D1 and D2 receptors (D1/D2), endothelial cells (Endo), ependymal cells (Epend), fast-spiking interneurons (FS), glutamatergic neurons (Glut), low-threshold-spiking inter-neurons (LTS), non-microglial macrophages (Macro), microglia (Micro), oligoden-drocytes (Oligo), oligodendrocyte progenitor cells (OPCs), and vascular smooth muscle cells (vSMCs). Light green-highlighted clusters denote non-neuronal cells, light orange-highlighted clusters denote medium spiny neurons, and light blue-highlighted clusters denote other neuronal populations. **b** Left, UMAP of 45,682 microglial cells, colored by subcluster. Right, dot plot of expression and prevalence of representative marker genes for each microglial subcluster. "Average Expres-sion" denotes the mean log-normalized expression level across cells in each sub-cluster, scaled for each gene, and "Percent Expressed" denotes the percentage of cells in which the log-normalized expression of the gene is greater than zero. **c** Scatter plot showing, for each of 129 individuals with at least 50 microglia cells, the percentage of inflammatory microglia and the subject's age (red, AUD; blue, no AUD). Bars on the left quantify the ratio of individuals with AUD to those without AUD among those with ≥50% (above the black line) or <50% (below) of microglia in the inflammatory state. **d** Left, UMAP of 130,129 astrocyte cells, colored by sub-cluster. Right, dot plot of expression and prevalence of representative marker genes for each subcluster. "Average Expression" and "Percent Expressed" are as described in (**b**). Source data are provided as a Source data file.

Individuals with at least half of their microglia in the inflammatory state were more likely to have AUD (70%) than those with fewer than half (52%) (Fig. 1c). There was also a significantly higher proportion in cluster 2 with increasing age (FDR = 0.029).

**Astrocyte clusters show distinct roles in synaptic maintenance and structural support**

A large astrocyte subcluster (Synaptic Astrocytes) was marked by higher expression of genes encoding excitatory amino acid

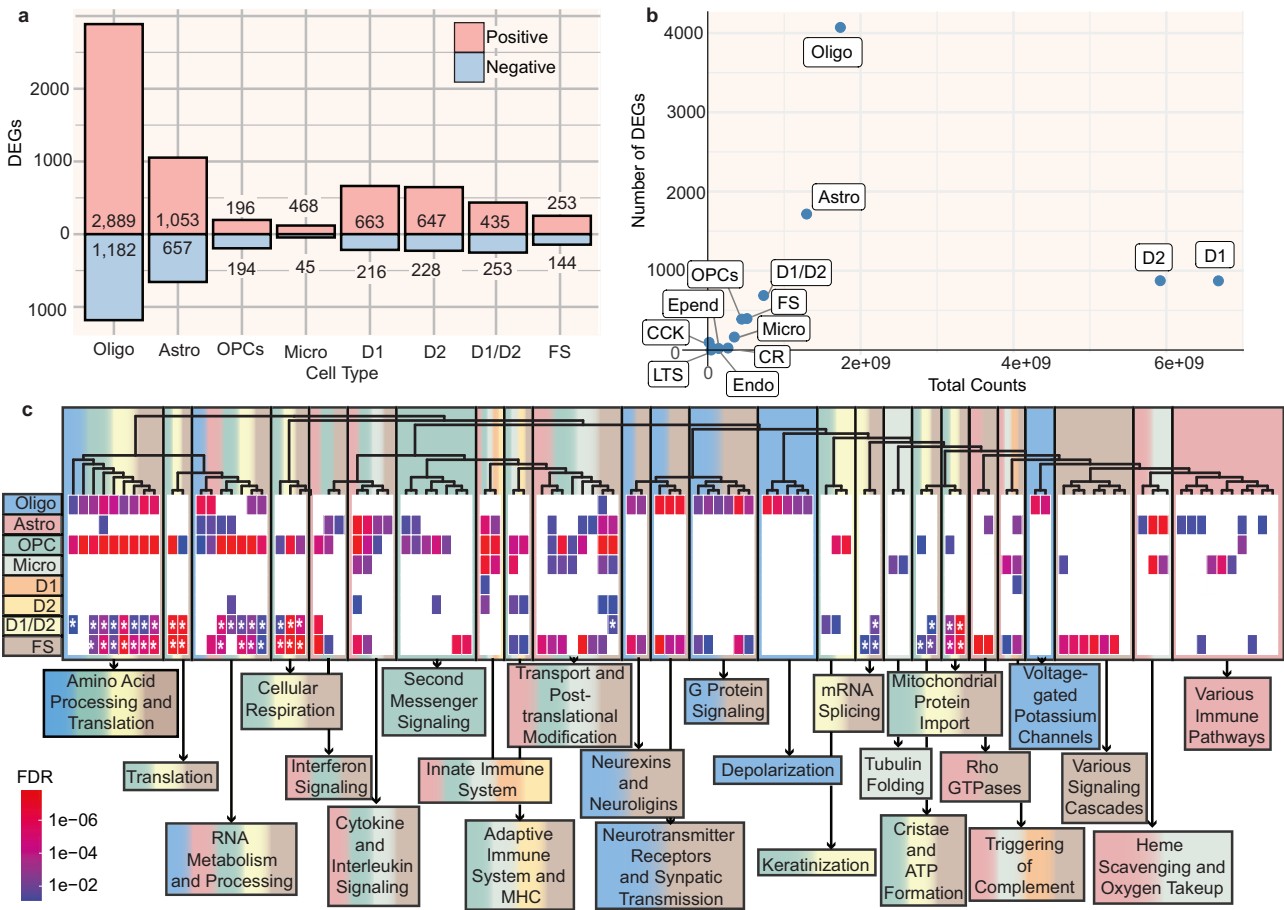

**Fig. 2 | Characterization of AUD-associated differences in gene expression in the caudate nucleus. a** Bar plot showing number of genes differentially expressed in individuals with AUD for the eight cell types which have over 100 differentially expressed genes (FDR < 0.05). Red and blue indicate positively and negatively differentially expressed genes (DEGs), respectively. DESeq2 was used for DEG testing. See Supplementary Data 7 for number of individuals tested for each cell type. Cell types: astrocytes (Astro), D1-type medium spiny neurons (D1), D2-type medium spiny neuron (D2), medium spiny neurons expressing both D1 and D2 receptors (D1/D2), endothelial cells (Endo), ependymal cells (Epend), fast-spiking interneurons (FS), microglia (Micro), oligodendrocytes (Oligo), oligodendrocyte progenitor cells (OPCs). **b** Total RNA-seq reads across individuals plotted against number of DEGs (FDR < 0.05) for each cell type, **c** Heatmap of biological pathways from the Reactome database enriched in brain samples from individuals with AUD in each cell type. The top 100 enriched pathways based on the smallest Benjamini–Hochberg adjusted *p* values (FDR) across all cell types are shown and were hierarchically clustered based on the number of genes shared between the pathways. Heatmap cell color indicates FDR. Asterisk indicates negative enrichment score; all other pathways have positive enrichment scores. The pathways are divided into 25 clusters, which are manually labeled with a brief summary of the pathways making up that cluster. Clusters are colored with a combination of colors corresponding to all cell types significantly enriched (FDR < 0.05) for at least 50% of the pathways in the cluster. Source data are provided as a Source data file.

transporters 1 and 2 (glutamate transporters), glutamate receptor 2 (an AMPA receptor subunit), and glutamate synthase, suggesting that these astrocytes may play a role in maintaining glutamatergic synapses (Fig. 1d, Supplementary Data 5 and 6). The other subcluster (Structural Astrocytes) was marked by high expression of cytoskeleton-related protein-coding genes *GFAP* and *DCLK1*, extracellular matrix protein tenascin C, and *CD44*, encoding a protein involved in cell adhesion and migration, and thus may be involved in structural support or tissue repair.

## Subclusters of medium spiny neurons correspond to anatomical compartments

There were two subclusters of both D1- and D2-type neurons, representing matrix and striosome compartments[37], based on expression of genes specific to either the matrix or striosome regions of the striatum[33] (Supplementary Fig. 4). 80% of D1 neurons and 83% of D2 neurons were within the matrix compartment, which makes up approximately 85% of the striatum[37]. There was no significant difference in subcluster proportion by AUD status in either astrocytes or D1- and D2-type neurons.

## Widespread differences in gene expression in diverse cell types are associated with AUD

We performed differential gene expression analyses in the thirteen major cell types in which there were greater than 50 cells in more than 10 individuals with and 10 without AUD. We used a pseudobulk approach, summing counts across cells within each sample for each cell type, with sex, age, and ethnic origin as covariates. Samples were removed on a cell type-specific basis if they contained fewer than 50 cells of that type (Supplementary Data 7). Eight cell types each contained over 100 DEGs (FDR < 0.05; Fig. 2a; Supplementary Data 7). We identified many changes in oligodendrocytes, astrocytes, and neurons, especially D1 and D2 MSNs. In each of these cell types, more of the DEGs had higher expression in individuals with AUD. Although the number of DEGs tended to be positively related to the number of reads in that cell type and the resulting higher statistical power, D1 and D2 MSNs had a markedly lower number of DEGs than other cell types with a similar number of reads (Fig. 2b). Many of the DEGs were differentially expressed in multiple cell types; notably, astrocytes and oligodendrocytes had 315 DEGs in common, and D1 and D2 neurons had 247 DEGs in common (Supplementary Fig. 5a).

## Biological pathways enriched in AUD vary by cell type, with immune response as a common theme

Gene set enrichment analysis using pathways from the Reactome database[39] showed that genes that differ in expression with respect to AUD were enriched in hundreds of pathways in many cell types (Fig. 2c, Supplementary Data 9). Successive hierarchical and manual grouping of pathways revealed that many immune response pathways—such as the adaptive immune system, innate immune system, and cytokine signaling in the immune system—were enriched in multiple cell types from individuals with AUD. In individuals with AUD, DEGs in oligodendrocytes were enriched for several pathways associated with synaptic regulation and depolarization, such as Neurotransmitter Receptors and Postsynaptic Signal Transmission and Voltage Gated Potassium Channels. D1/D2 and FS neurons had decreases in gene expression within pathways related to translation and metabolism, notably different than the canonical D1 and D2 MSN clusters, which had increases in gene expression mostly in immune-related pathways.

## AUD is associated with a net increase in chromatin accessibility in major cell types

We performed peak calling on the aggregated ATAC-seq fragments from each cell type, thereby identifying cell type-specific open chromatin regions (see Supplementary Information). Chromatin accessibility showed a strong peak near the TSSs of several cell-type-specific genes (Supplementary Fig. 6a, b). We identified differentially accessible chromatin regions (DARs; open chromatin regions that differed in accessibility between individuals with and without AUD) for eight cell types (Supplementary Data 12). Only oligodendrocytes, astrocytes, D1 neurons, and D2 neurons had over 15 DARs (FDR < 0.05; Fig. 3a, b). Just as with the DEGs, most of the differences were in the positive direction; chromatin was on average more open in samples from individuals with AUD. However, most of these differences in chromatin accessibility were relatively small; only in oligodendrocytes did any DARs surpass an absolute $\log_2$ fold change of 0.5.

To understand how differences in AUD-associated chromatin accessibility might influence gene expression, we compared the magnitude and direction of differences chromatin accessibility to the differences in gene expression for genes that had at least 1 DAR (FDR < 0.2) in the promoter region, a total of 4915 genes across all cell types. The AUD-associated DARs and DEGs were in the same direction for most genes in the four largest cell clusters (88%, 90%, 73%, and 77% in oligodendrocytes, astrocytes, D1 neurons, and D2 neurons, respectively). Genes containing DARs were enriched among DEGs in the same four cell types (FDR <1e-8; Fig. 3c–f). These results suggest that AUD-associated differences in chromatin accessibility can potentially lead to a corresponding change in *cis*-gene expression.

## Alcohol-related genes are enriched in key trans-regulatory modules in AUD

To determine which transcription factors and their target genes become more or less active in individuals with AUD, we used LINGER[40], a recently developed tool for inferring gene regulatory networks from paired single-cell expression and chromatin accessibility data (see "Methods"). We used the same pseudobulk expression and accessibility data used in the differential analyses above to construct the regulatory network and extracted key transcription factor-target gene subnetworks (modules) within it (Supplementary Data 14). Several regulatory modules were significantly enriched for genes from the two large-scale alcohol-related GWAS[4,6]. Modules 1, 2, 3, and 10 were enriched for genes associated with PAU, and modules 3 and 8 were enriched for genes associated with drinks per week (Supplementary Fig. 7a).

## FOXO1 is a key regulator of AUD-related expression differences in D2 MSNs

We identified several key regulatory transcription factors in each cell type using the expression and accessibility of all target genes within the trans-regulatory network constructed by LINGER[40] (Supplementary Fig. 7d, e). For example, in D2 MSNs, the transcription factor predicted to have the most significant difference between those with and without AUD, based on the chromatin openness of its target genes, was *FOXO1* (Forkhead transcriptional factor O1, Supplementary Fig. 7d), which has been previously shown to regulate energy homeostasis in neurons[41] and was linked to depression in recent studies[42,43].

## bZIP transcription factors are implicated in increased astrocyte reactivity in AUD

In astrocytes, LINGER identified six transcription factors implicated in regulating AUD-associated differences in both expression and chromatin accessibility of their target genes. These transcription factors included *JUND* (JunD proto-oncogene), which had increased regulatory activity in individuals with AUD (Supplementary Fig. 7d, e).

Transcription factors with increased activity in AUD plausibly impact the accessibility of related DNA motifs on their target genes[44]. We found 99 transcription factor motifs with significantly different enrichment in those with AUD (FDR < 0.2) using chromVAR[45]. Notably, 15 of the top 20 enriched motifs are annotated by JASPAR[46] as bZIP transcription factors (Supplementary Data 16), including *JUND* (FDR = 0.16), the only gene identified in both the gene regulatory network and motif enrichment analyses as having increased activity in astrocytes in AUD. An ANOVA using pseudobulk-level motif enrichment (see "Methods") confirmed a significant difference in *JUND* enrichment in individuals with AUD ($p = 0.0028$, Fig. 4a). Astrocytes with high *JUND* motif enrichment were largely in regions with high complement component 3 (*C3*) expression, a marker of reactive astrogliosis (Fig. 4b)[47]. *C3* expression was significantly higher in individuals with AUD (1.65 fold, FDR = 0.0007; Fig. 4c).

## OLIG2 is linked to dysregulation of myelin-related gene in AUD in oligodendrocytes

We saw a modest decrease in expression of myelin basic protein (*MBP*), a major component of myelin, in oligodendrocytes from individuals with AUD (FDR = 0.11). Graph-based subclustering of oligodendrocytes revealed three major subclusters, one of which had significantly lower *MBP* expression than the other two (Fig. 4d). This cluster was marked by high expression of *OLIG2*. We found 669 motifs with a significant difference in enrichment in those with AUD (FDR < 0.2), including *OLIG2* (FDR = 0.04, Supplementary Data 17). A pseudobulk-level ANOVA confirmed a modest enrichment ($p = 0.12$, Fig. 4e). This gene was identified as a "driver" by LINGER, having higher regulatory activity in individuals with AUD based on the chromatin accessibility of its target genes.

See Supplementary Information and Supplementary Fig. 7 for additional results from the gene regulatory network analysis.

## Activated microglia induce reactive astrocytes and oligodendrocytes

We measured changes in ligand-receptor signaling pathways between microglia, astrocytes, and oligodendrocyte cells using MultiNicheNet[48]. We identified three ligand-receptor pairs that signal from microglia to astrocytes with high downstream gene activity: *IL1B-IL1R1*, *OSM-OSMR*, and *TNF-TNFRSF1A* (Fig. 4f, see Supplementary Data 18 for all ligand-receptor pairs). These three pairs have been shown to work synergistically to induce pro-inflammatory cytokines in astrocytes[49]. Predicted downstream target genes of *IL1B-IL1R1* in astrocytes included *C3*, the widely used marker of reactive astrocytes, as well as bZIP family TFs *FOSL1*, *XBP1*, and *CEBPD*. We identified a single ligand-receptor pair from both astrocytes and microglia to

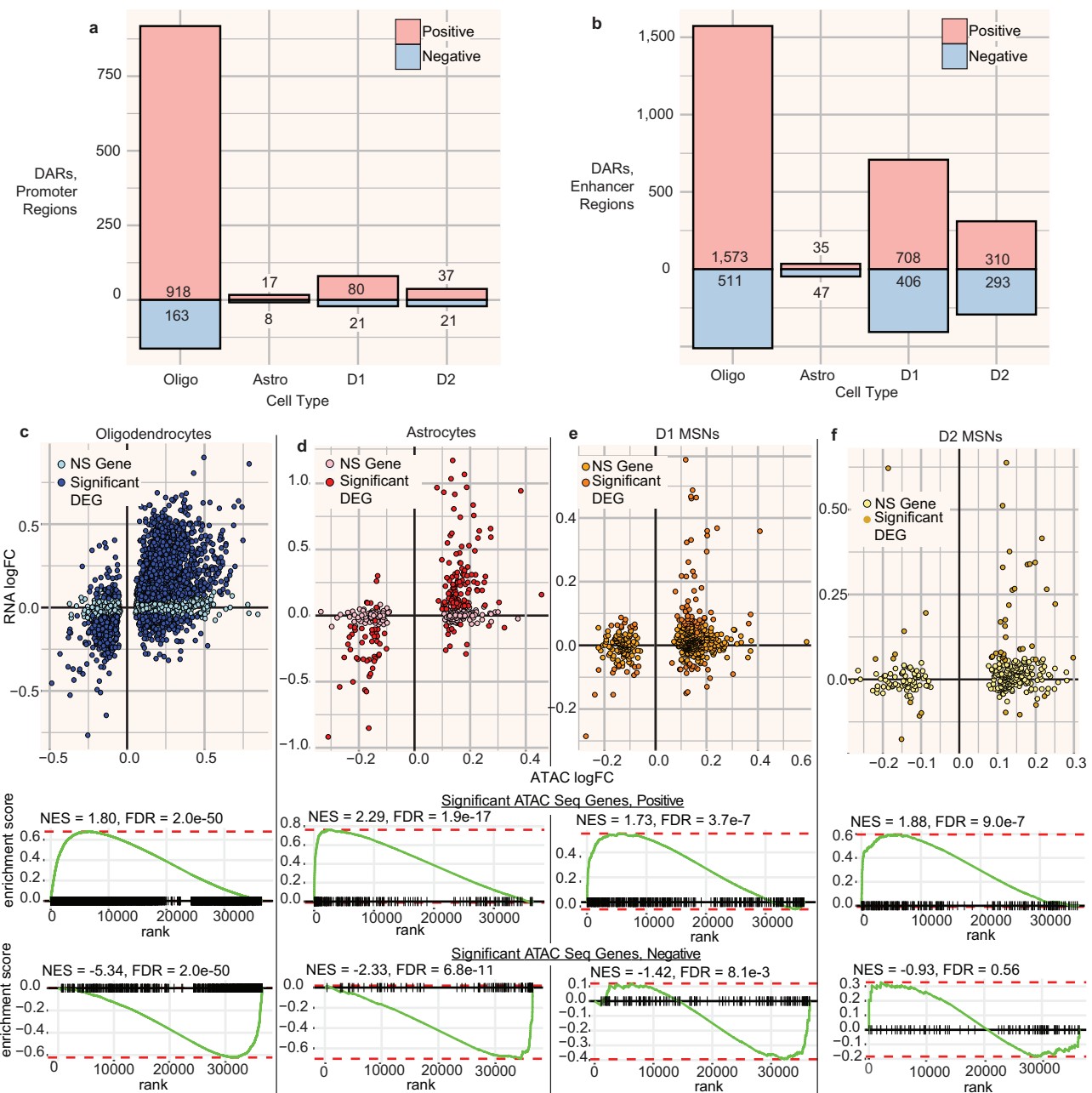

**Fig. 3 | Characterization of AUD-associated differences in chromatin accessibility in the caudate nucleus. a** Number of differentially accessible regions identified in oligodendrocytes, astrocytes, D1, and D2-type MSNs; red and blue indicate positively and negatively differentially accessible regions (FDR < 0.05), respectively, and lighter and darker coloring indicate regions in promoter regions of genes, respectively. Promoter regions are defined as a 1 kilobase region on either side of the transcription start side of each gene. See Supplementary Data 12 for the number of individuals tested for each cell type. **b** As a, for enhancer regions (>1 kilobase from the transcription start site of a gene). **c**−**f** Top, scatter plot of ATAC peak log fold changes and RNA-seq log fold changes for genes with at least one differentially accessible region in the promoter region (within one kilobase from the transcription start site) (FDR < 0.2, see below for number of genes plotted for each cell type). Genes are colored based on whether the gene is also differentially expressed (FDR < 0.2); Bottom, GSEA enrichment plot of enrichment of the same ATAC-significant genes, split into two sets based on positive or negative effect size, across genes ranked by differential expression fold change. Normalized enrichment score (NES) and Benjamini−Hochberg adjusted p value (FDR) for each GSEA test are shown. **c** oligodendrocytes, 4314 genes plotted; **d** astrocytes, 336 genes plotted; **e** D1 MSNs, 529 genes plotted; **f** D2 MSNs, 290 genes plotted. Source data are provided as a Source data file.

oligodendrocytes with a high ligand activity: *TGFB1-ITGB8* (Supplementary Fig. 8).

## Discussion

In this study, we present a large-scale integrated profile at the single-nuclei level of differences between individuals with and without AUD in gene expression, chromatin accessibility, and cell state in the caudate nucleus, and determined potential regulatory mechanisms underlying these differences. By combining snRNA-seq with chromatin accessibility profiling (ATAC-seq) within the same cells at large scale, using both sn-multiome and snRNA-seq experiments, we discovered unique patterns of gene expression within different cell types and distinct regulatory mechanisms underlying them. This integrative approach demonstrates the power of large-scale multiomic studies to uncover cell-type-specific regulatory mechanisms in complex brain disorders, and can be applied to investigate other neuropsychiatric and neurodegenerative conditions.

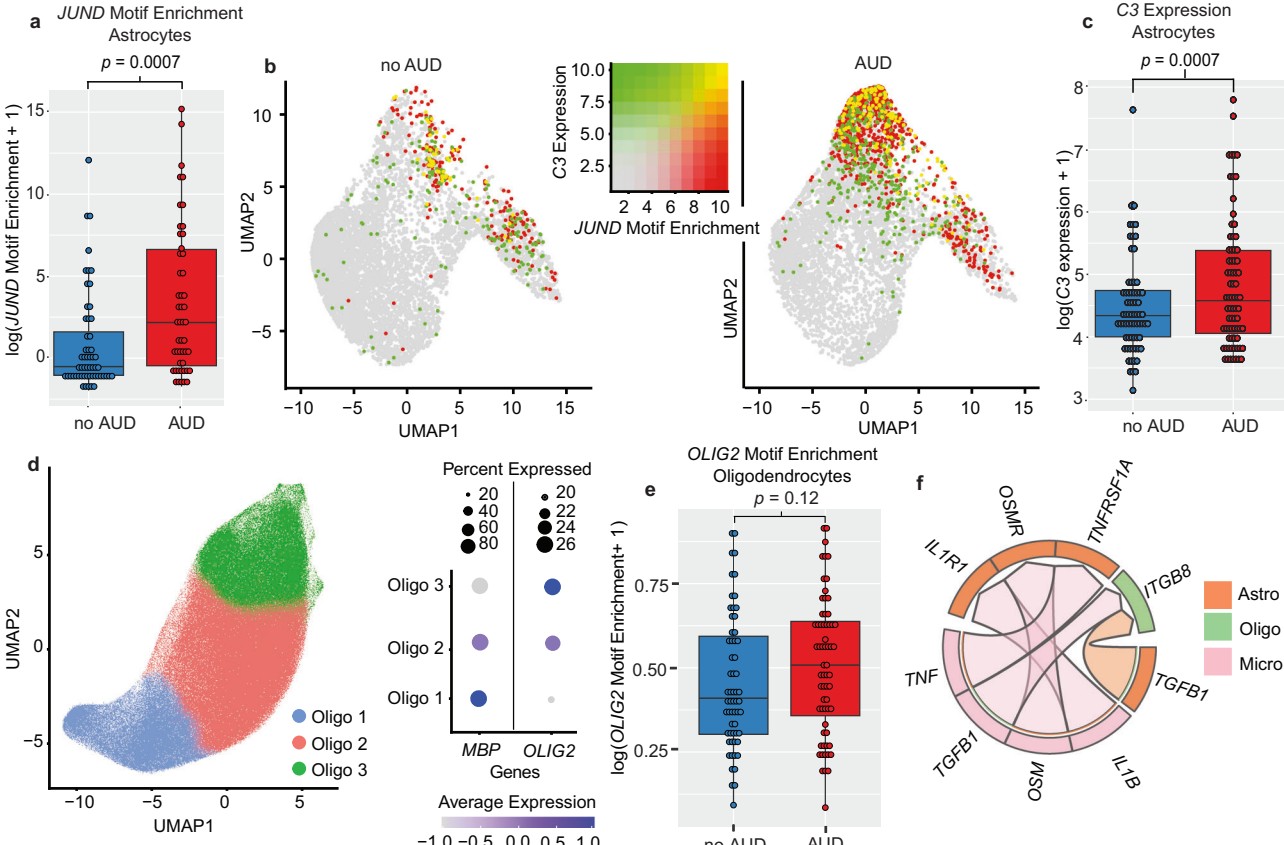

**Fig. 4 | Cell type-specific gene regulatory networks associated with AUD.**
**a** Boxplot of the log of chromVAR motif activity score for the *JUND* motif in astrocytes for samples from 105 individuals with and without AUD (49 AUD, 56 no AUD). Center of boxes denote median log Enrichment Score, with box boundaries denoting Q1 (25th percentile) and Q3 (75th Percentile). Bottom whisker edge denotes Q1−1.5 × Interquartile Range (IQR), and top whisker denotes Q3 + 1.5 × IQR. Here and following, "log" denotes natural logarithm. *P* value is reported from a Wilcoxon Signed-Rank Test for the difference in *JUND* motif activity between individuals with and without AUD. **b** UMAP of 13,911 astrocyte cells from individuals without AUD (left plot) and with AUD (right plot). Dot color indicates the enrichment of the *JUND* motif (red) and the log-normalized *C3* expression (green). Yellow indicates high expression of *C3* and high *JUND* motif enrichment. **c** Boxplot of the log of *C3* expression in astrocytes for samples from 143 individuals with and without AUD (74 AUD, 69 no AUD). Box center, edges, and whiskers are defined as in (**a**). *P*

value is reported from a Wilcoxon Signed-Rank Test, as in (**a**). **d** Left, UMAP of 325,593 oligodendrocyte cells, clustered and annotated into three subclusters using graph-based clustering. Right, dot plot of *MBP* and *OLIG2* expression and prevalence for each oligodendrocyte subcluster. "Average Expression" denotes the mean log-normalized expression level across cells in each subcluster, scaled for each gene, and "Percent Expressed" denotes the percentage of cells in which the log-normalized expression of the gene is greater than zero. **e** Boxplot of the log of chromVAR motif activity score for the *OLIG2* motif in oligodendrocytes for samples from 124 individuals with and without AUD (64 AUD, 60 no AUD). Box center, edges, and whiskers are defined as in (**a**). The *p* value is reported from a Wilcoxon Signed-Rank Test, as in (**a**). **f** Circos plot showing the top five ligand-receptor interactions (determined by scaled ligand activity score from MultiNicheNet) between astrocytes, oligodendrocytes, and microglia. Colors denote cell types, and arrows denote direction of signaling. Source data are provided as a Source data file.

Our large sample size allowed us to identify and study 17 distinct cell types in the human caudate nucleus; most correspond well to a recent basal ganglia cell-type atlas of the human brain[50]. We identified rare interneuron clusters, as well as microglia and oligodendrocytes subtypes not identified in that atlas. Our large caudate-specific atlas sheds more light on the two astrocyte subtypes identified therein.

We identified thousands of genes differentially expressed in different cell types and characterized the accompanying differences in chromatin accessibility. We observed an overall increase in both gene expression and chromatin accessibility associated with AUD, which may reflect a heightened state of cellular activity in response to chronic alcohol exposure. Chronic alcohol use has been shown to induce widespread neuroadaptive changes, including alterations in synaptic plasticity in the rodent dorsal striatum[51], inflammatory signaling in the human and rat hippocampus and prefrontal cortex[24,52], and cellular stress responses, especially along the hypothalamic-pituitary-adrenocortical axis[52,53]. Here, in our human caudate dataset, we indeed observed differences in enrichment of pathways relating to these processes—such as cytokine/interferon response,

innate immune system, and complement cascades—in every cell type that we tested. These observations may therefore represent a composite of responses that emerge as a result of chronic alcohol exposure.

Pathways related to RNA processing ("RNA Metabolism," "Processing of Capped Intron-Containing pre-mRNA") and immune response ("Innate Immune System," complement-related pathways) were enriched in D1- and D2-type MSNs (the projection neurons of the basal ganglia) in individuals with AUD. Gene regulatory network analysis allowed us to identify transcription factors that may regulate these differences. For example, *FOXO1* was predicted to have significantly lower activity in D2 MSNs from individuals with AUD, based on the chromatin openness of its target genes. In mice, *FoxO1* has been linked to the regulation of energy homeostasis in neurons[41], and a lack of *FoxO1* in the brain caused a depressive-like phenotype[42,43]. This might be relevant to the frequent co-occurrence of AUD and major depression[54]. The finding that different regulatory modules were enriched for genes associated with PAU vs. AUD further reinforces the genetic differences between those traits[8,55].

We observed a small cluster of MSNs expressing both D1 and D2 dopamine receptor genes, variously described as eccentric MSNs, D1H, and D1/D2 hybrid MSNs[33,56], which have been observed in mice[56,57], primates[33], and humans[58]. These neurons, which showed strong enrichment of the marker *RXFP1*, showed a markedly different gene expression profile than that of in silico-created doublets of D1 and D2 cells. These neurons have been shown in a recent study in mice to be morphologically distinct from D1- and D2-type MSNs, with a smaller cell body, less expansive dendrite structure, and fewer spines, and were differently affected by treatment with a denervating agent[57]. The pattern of gene expression changes and pathways in these neurons differed from those of the other MSN types, suggesting that D1/D2 hybrid MSNs play a distinct role in the caudate of those with AUD.

In microglia, we identified four distinct states that did not adhere closely to the classical M1/M2 (inflammatory/anti-inflammatory) distinction[35]. Indeed, recent studies are beginning to question the biological accuracy of the widely-cited M1/M2 classification[59]. An increased proportion of microglia showed an inflammatory gene expression profile in those with AUD. Chronic alcohol exposure has been shown to cause microglial activation in mice, leading to neuroinflammation[60]. In microglia, we identified a sole regulatory module driven by *ZBTB16*, a negative regulator of inflammation[61], that exhibited reduced expression in individuals with AUD. Therefore, this decrease in *ZBTB16*-mediated regulation may further exacerbate inflammatory responses in microglia in AUD.

In astrocytes from individuals with AUD, there was significantly higher expression of the astrocyte reactivity marker *C3*. This extends to the human caudate evidence that inflammation evoked by ethanol exposure is accompanied by reactive astrogliosis in the nucleus accumbens of rats[13]. A subset of astrocytes with increased *C3* expression had significantly higher motif enrichment of bZIP transcription factors like *JUND*. Gene regulatory network analysis also implicated increased *JUND* in AUD, suggesting that changes in these factors may be regulating astrocytes as they undergo astrogliosis.

In oligodendrocytes, we observed differences in expression and chromatin accessibility in thousands of genes, and enrichment in biological pathways relating to neurotransmitter uptake and depolarization. The gene encoding myelin basic protein was expressed at lower levels in individuals with AUD only in cells with higher expression of *OLIG2* (Fig. 4g, h). *OLIG2*, a master regulator in mature and developing oligodendrocytes, has been shown to have higher activity after brain injury[62] and is linked to myelination: replacing *Olig2* with its dominant-active form in rodents led to decreased expression of *MBP*[63], and deletion of the *Olig2* gene accelerated remyelinating processes[64]. This suggests that our observed increase in *OLIG2* activity in individuals with AUD may, in part, lead to dysregulation of myelination in oligodendrocytes. Indeed, alcohol consumption and AUD have been found to be associated with white matter degeneration[65,66], but prior to this study, there was no direct link between AUD, demyelination, and specific genes such as *OLIG2*.

We integrated the transcriptomic and epigenetic alterations in each cell type into predictions of upstream signaling events. We found increased signaling via *TFGB1-ITGB8* from both microglia and astrocytes to oligodendrocytes. TGF-$\beta$1 signaling is known to increase after injury, and studies have shown that ethanol exposure induces TGF-$\beta$1 signaling in rats[67,68]. Signaling from microglial cells to astrocytes that involves pro-inflammatory molecules IL-1$\beta$, TNF, and oncostatin M is higher in individuals with AUD, concordant with the hypothesis that activated microglia induce neurotoxic reactive astrocytes[47]. These three molecules work synergistically in astrocytes and other cells to induce pro-inflammatory and neurotoxic molecules such as nitric oxide[69] and prostaglandin E(2)[49]. Although reactive astrocytes can induce death of neurons and oligodendrocytes, we did not observe a significant difference between individuals with and without AUD in relative proportions of neuronal cell types or oligodendrocytes.

There are some caveats to our analyses. One limitation is that those who drink heavily are more likely to smoke[70]. Thus, some differences might be attributed in part to smoking. Nevertheless, these findings remain a valid reflection of the disease's underlying biology, which often includes smoking. Another limitation is that our results are primarily from males genetically closely related to the 1000 Genomes European samples, and therefore do not capture the transcriptional and epigenetic diversity across ancestry groups or sexes[71]. AUD is more likely to occur in men, and our cohort demographics reflect this; thus, we did not have adequate statistical power for sex-specific analysis. Differential analysis did show some genes whose expression was significantly associated with sex (e.g., in astrocytes, 186 sex associated genes were identified with FDR < 0.05; age, ethnic origin, and AUD status as covariates). There were many gene expression differences associated with age (e.g., in astrocytes, 1908 age-associated genes were identified with FDR < 0.05; age, ethnic origin, and AUD status as covariates). However, since the AUD and non-AUD groups were age-matched (AUD average age = 54.9; non-AUD average age = 56.7) and age was included as a covariate in the above analyses, this does not impact our ability to identify patterns of expression and chromatin differences associated with AUD.

Another limitation is that in post-mortem brain samples, both pre-existing differences that increase risk for AUD and differences associated with the extended, excessive alcohol consumption characteristic of AUD are present. Some AUD-associated differences we observed (e.g., changes in inflammatory and myelinating processes) are likely to be associated with the consequences of AUD. Therefore, future studies should further leverage genetic information to tease apart cause and consequence. Future directions should also include experimental validation of the upstream genes identified in the above gene regulatory network and cell-cell communication computational analyses.

In conclusion, we provide a detailed picture of the vast transcriptional and epigenetic differences between individuals with and without AUD in many different cell types within the caudate nucleus that illuminates biological mechanisms underlying these differences and identifies potential driver genes causing these differences. Our work provides important findings into the etiology associated with AUD, pointing to key pathways and regulatory genes, and underscores the potential of large-scale multiomic datasets to provide meaningful insights into brain regions and diseases.

## Methods

### Sample collection

The caudate from post-mortem brains of 183 donors were initially included in this study. Tissue was obtained from the New South Wales Brain Tissue Resource Centre (NSWBTRC), University of Sydney, Australia (https://sydney.edu.au/nsw-brain-tissue-resource-centre)[72].

### Genotype data processing and imputation

NSWBTRC samples were genotyped using the UK Biobank Axiom® Array (ThermoFisher Scientific, Waltham, MA, U.S.A.). Before imputation, palindromic SNPs and SNPs with genotyping rate <95%, minor allele frequency < 1%, or Hardy-Weinberg equilibrium *P* value < 1E-4 were excluded. Genotype data were imputed by using the TOPMed Imputation Server[73]. Eagle v2[74] was used to phase the genotypes, and Minimac4 v1.2.4[73] was used for imputation. Data from the Trans-Omics for Precision Medicine (TOPMed r3)[75] were used as the reference genomes. Variants with imputation $R^2 > 0.995$ were retained for use in demultiplexing (see below).

### Single-cell multiome assay

**Nuclei isolation for single-cell multiome.** 183 fresh-frozen post-mortem caudate brain samples were divided into 23 pools, with 8 in each pool. The donors in each pool were both condition (with or without AUD) and sex balanced. For each pool, around 20 mg tissue

from each donor specimen was collected and combined into a sterilized 2 ml Dounce homogenizer. 2 ml chilled NP40 lysis buffer (10 mM Tris-HCl, pH 7.5, 10 mM NaCl, 3 mM MgCl2, 0.1% Nonidet P40 Substitute, 1 mM DTT, 1 U/µl RNase inhibitor) was added to the Dounce homogenizer before the tissues were thawed. The tissues were homogenized 15× using pestle A, and 10× using pestle B, and were transferred into a centrifuge tube to incubate for 2 min on ice. After that, 2 ml wash buffer containing PBS, 1% BSA, and 1 U/µl RNase inhibitor was added and mixed well. The lysed tissue was centrifuged at 500 × $g$ for 5 min at 4 °C, then washed twice more with wash buffer and filtered through 70 µm and then 40 µm cell strainer separately. The pellet was resuspended in 2 ml wash buffer and mixed with 3.6 ml Sucrose Cushion Buffer I (nuclei PURE prep isolation kit, Sigma) containing 1 U/µl RNase inhibitor. Two milliliters of Sucrose Cushion Buffer I with 1 U/µl RNase inhibitor was added into one 15 ml Beckman Coulter centrifuge tube. After that, the 5.6 ml nuclei suspensions were gently added to the top of Sucrose Cushion Buffer I without mixing, followed by centrifuging at 13,000 × $g$ (Beckman Coulter ultracentrifuge) with rotor SW40Ti for 45 min at 4 °C. The purified nuclei pellet was washed by centrifuging at 300 × $g$ for 5 min at 4 °C with wash buffer, and the washed nuclei pellets were resuspended in wash buffer to target ~1000 nuclei/µl.

**10X single-cell multiome library preparation and sequencing.** Paired ATAC and gene expression libraries were generated following the Chromium Next GEM Single Cell Multiome ATAC + Gene Expression User Guide CG000338_RevB (10X Genomics, Inc). In brief, the isolated nuclei from a pool of samples were first incubated in a transposition mix. The single nuclei master mixture containing tagmented single nuclei suspension was loaded into two wells of a Next GEM Chip J, along with the single cell multiome gel beads and partition oil. The chip was then loaded into the Chromium X Controller for GEM generation and barcoding. Barcoded transposed DNA and cDNA were amplified after the GEMs were released. At each step, the quality of the cDNA, ATAC library, and cDNA library was examined by Bioanalyzer 2000. The final single-indexed ATAC libraries and the dual-indexed gene expression libraries were sequenced on an Illumina Novaseq 6000, with index reads of 10 bp + 24 bp, and 100 bp paired-end reads.

**Cell Ranger ARC analysis.** Cell Ranger ARC (cellranger-arc-2.0.0, http://support.10xgenomics.com/) was utilized to process the ATAC and gene expression FASTQ files derived from the single-cell multiome libraries. The reference refdata-cellranger-arc-GRCh38-2020-A-2.0.0 (10x Genomics) was used. The filtered gene-cell barcode matrices and fragment files were used for further analysis.

### Single-nuclei RNA-seq assay
**Nuclei isolation for single-nuclei RNA-seq.** 170 fresh-frozen postmortem caudate brain samples (same individuals as in the multiome assay) were grouped into 17 pools, with 10 in each pool. The donors in each pool were both condition (with or without AUD) and sex balanced. The nuclei isolation for each pool is similar to that described for the single cell multiome assay, above. Zero point two units per microliter of RNase inhibitor was used in the buffers.

**10X HT single-nuclei RNA-seq library preparation and sequencing.** The Chromium Next GEM single cell 3′ HT reagent kits v3.1 (user guide CG000416, 10X Genomics, Inc.) were used for the single-nuclei RNA-seq assay. The single nuclei suspension from a pool of 10 donor tissue samples was loaded into two wells of a Chromium Next GEM chip M to target 60,000 cell recovery per well. The chip was run on a Chromium X (10x Genomics). Single-cell gel beads in emulsion containing barcoded oligonucleotides and reverse transcriptase reagents were generated. cDNA was synthesized and amplified following cell capture and

cell lysis. The quality and quantity of cDNA and resulting libraries were examined by Bioanalyzer. The final libraries were sequenced on an Illumina NovaSeq 6000. 100-bp reads, including cell barcode and UMI sequences, and 100-bp RNA reads were generated.

**Cell Ranger Count analysis.** Cell Ranger Count (cellranger-count-7.0.1, http://support.10xgenomics.com/) was utilized to process the FASTQ files derived from the snRNA-seq libraries. The reference refdata-cellranger-GRCh38-2020-A (10x Genomics) was used. The filtered gene-cell barcode matrices were used for further analysis.

### Demultiplexing
Cells from each of the 40 sequencing pools (23 from the single-cell multiome assay and 17 from the single-nuclei RNA-seq assay) were demultiplexed back into their samples of origin using the tool Demuxlet[76] with default parameters, using the genotype array described above for the reference genotypes of each individual. Between 55% and 75% of cells from each pool were identified as singlets and assigned to a sample. The remaining cells (identified as doublets or ambiguous) were removed from further analysis.

After demultiplexing, seven samples from the HT assay and twenty-three samples from the multiome assay with either no genotype info available or less than 100 cell barcodes assigned were removed from all further analyses. Raw data from the barcodes from these samples are publicly available, see "Data availability," below. These samples—163 from the HT experiments, and 161 from multiome—were used for all the following processing and analyses until the filtering step detailed in "Sample filtering," below.

### Initial quality control
Unless specified differently, all following analysis was performed in R (version 4.3.1), predominantly utilizing the Seurat[77] (v5.0.0) and Signac[78] (v1.12.0) packages.

A Seurat object was created from the data from each sequencing pool from the HT assay using the gene expression count matrix from the Cell Ranger output. Cells with below 800 or above 11,250 genes, above 125,000 molecules, or above 10% mitochondrial RNA were removed from further analysis (Supplementary Data 19). Each pool was then normalized using the scTransform() function.

A Signac object, containing both RNA and ATAC-seq data, was created for each pool from the multiome assay using the HDF5 file from the Cell Ranger output. Cells with below 800 or above 20,000 genes, below 800 or above 500,000 detected RNA molecules, above 20% mitochondrial RNA, or above 8% ribosomal RNA were removed from further analysis. An additional round of filtering was performed using the ATAC-seq data. The following cells were removed from further analysis (Supplementary Data 20):

- Cells with less than 100 or over 100,000 features;
- Less than 100 or over 1,000,000 counts;
- TSS enrichment less than 2;
- Nucleosome signal greater than 4;
- Percentage of reads in peaks less than 15%;
- Total number of fragments in peaks less than 800 or over 100,000;
- Ratio reads in genomic blacklist regions greater than 0.05.

Between both assays, 1,307,323 unique barcodes passed all QC filters.

### RNA-seq integration and visualization
After the above quality control, all cells from the Seurat objects for each pool were integrated for visualization in the same 2D space. Atomic sketch integration was used, a dictionary learning based procedure for large datasets (see https://satijalab.org/seurat/articles/parsebio_sketch_integration). Briefly, 5000 representative cells were

selected from each pool (based on statistical leverage). Integration was performed on these sketched cells using the RPCAIntegration method. Then, each cell was placed in this integrated space as well using the ProjectIntegration function.

To visualize all cells in the same plot, we converted each pool to an on-disk BPCells matrix[79]. This allowed us to merge each object in a memory-efficient way. After merging, the function RunUMAP was run on the combined object for 2D visualization.

## Cell type annotation

The 1,307,323 cells were divided into 49 clusters using the FindNeighbors and FindClusters functions in Seurat. Cell clusters were annotated into known striatal cell types based on expression levels of a combination of marker genes curated from established studies (Fig. 2b)[30,31,33,56,80]. Cells were mapped to the Allen Institute Mammalian Basal Ganglia Consensus Cell Type Atlas[34] for validation using the cell_type_mapper Python package provided by the Allen Institute.

## Cell type abundance

scCODA[81] was used to test for relative differences in abundance of cell clusters between AUD and non-AUD groups, using an FDR cutoff of 0.05. A nonzero "final parameter" statistic for a given cell type indicates a statistically significant difference in abundance.

## ATAC-seq integration, visualization

Peaks from each of the Signac objects were merged to create a common peak set. Each of the 23 objects was then processed using the standard ATAC-seq procedure in Signac—FindTopFeatures, RunTFIDF, RunSVD—and all objects were then integrated using the IntegrateEmbeddings command.

Cell type labels were transferred to the ATAC-seq object using the assignments for each barcode determined from the snRNA-seq data. Barcodes without a matching QC-passed snRNA-seq barcode were excluded, leaving 250,537 cells from 159 samples. UMAP visualization was calculated with the RunUMAP command, using the integrated_lsi reduction.

## Sample filtering

Following the above analyses, 20 samples with a proportion of glutamatergic neurons greater than 10% were removed from both the processed RNA-seq and ATAC-seq data, because such a cell-type composition indicates potential contamination with non-caudate tissue, leaving 143 samples for the following downstream analyses.

## Cell subtype annotation and testing

Microglia and astrocyte clusters were further divided into four and two subclusters, respectively, by performing another iteration of FindNeighbors and FindClusters, using cells from the 143 samples. Subcluster-specific genes were determined by using the roc test within the FindMarkers function in Seurat (see https://satijalab.org/seurat/reference/findmarkers). The top 50 genes (based on myAUC statistic) were used as input into g:Profiler[82] to determine enriched biological pathways specific to that subcluster. (Notably, g:Profiler uses their own method, called gSCS, for multiple testing correction in lieu of Benjamini–Hochberg (BH) method)

For testing for a difference in the proportion of cell states in individuals with AUD, the mean proportion of cells in each cluster was calculated for each sample, and an ANOVA was performed to determine if the mean proportion significantly changed in samples from individuals with AUD as compared to those without. Age, sex, and ethnic origin were used as covariates. For this test, we removed samples with fewer than 50 cells of the cell type being tested. FDR correction of p values was performed using the BH procedure, as described in their original paper[83], using an FDR cutoff of less than 0.05.

## Differential expression analysis

**RNA pseudobulk samples creation.** Due to the sparsity of single-cell data, differential expression methods designed to be run on the single-cell level often lack high statistical power. To account for this challenge, we utilized a pseudobulk approach. To create the pseudobulk data, for each cell type, the gene expression matrices of each cell of that cell type were summed by sample ID. Samples were removed on a cell type-specific basis if the sample contained less than 50 cells of that cell type (Supplementary Data 7). Due to a low number of samples (less than 10 individuals with AUD and 10 without) meeting the ≥50 cell criteria, differential gene expression analysis was not performed for cholinergic interneurons, vascular smooth muscle cells, CCK interneurons, and macrophages. Supplementary Figs. 9–11 display samples plotted by the top two principal components of pseudobulk-level expression for each cell type, separated by AUD classification and sex.

**Differential gene expression analysis.** Differential gene expression analysis between samples from individuals with and without AUD was performed for each cell type, using DESeq2[84], with the default parameters. Sex, age (as a continuous variable), and ethnic origin were included as covariates in the models. Genes with FDR of less than 0.05 (corrected for multiple-hypothesis testing using the BH method) were deemed significant.

## Gene set enrichment analysis

Gene set enrichment analysis was performed for each cell type that underwent differential expression analysis, using the fgsea R package[85]. The $\log_2$ fold changes from the differential expression analysis were used as the ranks, and pathways from the Reactome database were used as gene sets (Supplementary Data 9, all pathways with FDR < 0.2 are shown). For visualization, the top 30 enriched pathways (based on the smallest FDR) in each cell type were selected and hierarchically clustered based on the number of genes shared between the pathways. Clustered pathways were then manually labeled into 25 groups. FDR correction of p values was performed using the BH procedure.

## Creation of cell type-specific ATAC-seq profiles

CoveragePlot function in Signac was used for the visualization of ATAC-seq signal for marker genes.

Peak calling was performed separately for cells from each of the 16 cell types (excluding glutamatergic neurons; as the caudate is not known to possess excitatory neurons, the presence of this cell type could indicate inadvertent inclusion of another brain region at the time of dissection) using the CallPeaks function in Signac with default parameters. The function uses MACS2[86] for peak calling.

All cells from the integrated ATAC-seq object, from the 143 samples determined after the "Sample Filtering" step, above, were used for the procedures in this section.

## Differential chromatin accessibility analysis

**ATAC pseudobulk samples creation.** In the same way as the RNA-seq data, pseudobulk chromatin accessibility data were created for the ATAC-seq data: For each cell type, the ATAC-seq count matrices of each cell of that cell type were summed by sample ID, for the 143 samples. Samples were removed on a cell type-specific basis if the sample contained less than 50 cells of that cell type (Supplementary Data 12). Due to a low number of samples meeting the ≥50 cell criteria, differential accessibility analysis was not performed for cholinergic interneurons, vSMCs, CCK interneurons, macrophages, ependymal cells, LTS interneurons, or endothelial cells. Supplementary Figs. 12 and 13 display samples plotted by the top two principal components of pseudobulk-level counts for each cell type, separated by AUD classification and sex.

**Differential accessibility analysis.** Differential chromatin accessibility analysis between individuals with and without AUD was performed for each cell type using DESeq2 with the default parameters. Sex, age, and ethnic origin were included as covariates. Genes with FDR < 0.05 (corrected for multiple-hypothesis testing using the BH method) were deemed significant. Regions residing in promoter regions of genes were determined using the R package ChIPSeeker[87]. The function annotatePeak() was used, with parameters: TxDb = TxDb.Hsapiens.UCSC.hg38.knownGene, annoDb = "org.Hs.eg.db," and tssRegion = c(−1000, 1000).

**Comparison of differentially accessible genes and differentially expressed genes.** To calculate the association between gene expression and chromatin accessibility differences for each gene, we assigned to each gene with at least 1 DAR (FDR < 0.2) in the promoter region the $\log_2$ fold change of the DAR with the highest ATAC signal, as well as the $\log_2$ fold change of the gene's expression.

For the GSEA analyses, the $\log_2$ fold changes from the differential expression analysis were used as the ranks, and genes with at least 1 DAR in the promoter region were used as the gene sets, separated into genes with positive $\log_2$ fold changes and those with negative $\log_2$ fold changes.

## Gene regulatory mechanisms prediction

**chromVAR[88].** Prediction of motif activities for each cell was performed using the RunChromVAR command in Signac[88]. Differential testing on the chromVAR z-score was performed using the FindMarkers function, setting mean.fxn=rowMeans and fc.name = "avg_diff," so that the fold change represents the average difference in z-score between the groups.

To make these differential activity motif results more robust, we also utilized a pseudobulk approach: averaging per-cell motif scores for all cells within a sample of a given cell type, taking the log, and then using an ANOVA—with sex, age, and ethnic origin as covariates—to test for differences between those with and without AUD. All samples used for differential accessibility testing (see "ATAC pseudobulk samples creation") were used for this analysis.

**Gene regulatory network inference.** To build the cell population gene regulatory network, we used LINGER[40]. We generated pseudobulk-level expression and chromatin accessibility data for each donor and each cell type. Here, we used the union set of peaks across all cell types (described above) for ATAC-seq data and sex, age, and ethnic origin as covariates to the model. All samples used for differential accessibility testing (see "ATAC pseudobulk samples creation") were used for this analysis.

**Identification of cis and trans driver TFs.** We identified *cis* and *trans* driver TFs underlying epigenetic and transcriptome changes between individuals with and without AUD using a linear regression model, $Y = A\beta + \beta_0 + \varepsilon$. For transcriptome drivers, the regression model predicted the log transformation of the gene expression fold change between individuals with and without AUD ($Y$) from cell type-specific TF-TG trans-regulation ($A$). For epigenetic drivers, the model predicted chromatin accessibility changes ($Y$) from the cell type-specific TF-regulatory element cis-regulation ($A$). Significant TFs from each model indicated TFs driving differential expression and chromatin states between conditions through direct epigenetic or transcriptome regulation. TFs with (BH corrected) FDR < 0.05 were deemed significant.

## Cell-cell communication analysis

To analyze cell-cell communication differences in individuals with AUD, we used MultiNicheNet[48]. All samples used for differential expression testing (see "RNA pseudobulk samples creation") were used for this analysis. User-set parameters were as follows:

- The minimum number of cells per cell type per sample was set to 10, the recommended default;
- Sex, ethnic origin, and age were used as covariates;
- For a DEG to be further considered when calculating ligand activity, we choose a minimum $\log_2$ fold change of 0.50, a maximum FDR of 0.2, and a minimum fraction of expression of 0.05;
- For the NicheNet ligand-target inference, the top 250 predicted target genes were considered;
- The weights of the prioritization of expression, differential expression, and NicheNet activity information were set to the recommended defaults (see https://github.com/saeyslab/multinichenetr);
- Sender cell types were defined as astrocytes, microglia, and oligodendrocytes;
- Receiver cell types were defined as astrocytes, microglia, and oligodendrocytes.

Visualization of the top 5 ligand-receptor pairs in individuals with AUD, based on scaled ligand activity score, was created using the make_circos_group_comparison function.

## Inclusion and ethics statement

This study was a collaborative, multi-disciplinary, multi-institutional effort with contributions from researchers in academic positions. All collaborators of this study have fulfilled the criteria for authorship required by Nature Portfolio, as their participation was essential for the design and implementation of the study. Roles and responsibilities were agreed upon among collaborators ahead of the research.

The NSWBTRC has ethics approval from the University of Sydney. The donor recruitment program (called the "Using our Brains" donor program) was established in 2002 as a pre-mortem consent program inviting members of the community (those living within the NSW Sydney Metropolitan, Hunter, or Illawarra Region) to donate their brain to neuroscience research after they die. Their collection focuses on controls, alcohol-related brain damage, and mental illness, specifically schizophrenia. The NSWBTRC encourages those affected with specific brain disorders and those who are medically healthy to donate. Those eligible to be a donor: (1) are aged 18 years and over, (2) live in NSW Sydney Metropolitan, Hunter or Illawarra Region, (3) are not a Whole Body Donor (different to organ donation), (4) do not have an infectious disease such as Hepatitis, HIV, AIDS, CJD, (5) have not been diagnosed with a brain tumor, stroke, epilepsy, or other neurological illness, (6) do not suffer from a serious head injury with loss of consciousness. Eligible donors complete a pre-screen via telephone or the online inquiry form and complete a "consent kit" that contains detailed information on the Using our Brains program and consent forms.

This study was not restricted to any settings of the researchers, and has minimized potential stigmatization, incrimination, and discrimination to all its readers. Local and regional research relevant to our study was taken into account in citations.

## Reporting summary

Further information on research design is available in the Nature Portfolio Reporting Summary linked to this article.

## Data availability

The GWAS datasets utilized in this study were obtained from: (1) Saunders et al.[4,6], summary statistics of which can be found at the Data Repository for the University of Minnesota [https://doi.org/10.13020/przg-dp88] (2) Zhou et al.[6], of which full summary-level information can be found at Yale School of Medicine [https://medicine.yale.edu/lab/gelernter/stats/] and dbGaP accession code phs001672 [https://www.ncbi.nlm.nih.gov/projects/gap/cgi-bin/study.cgi?study_id=phs001672]. The data generated here, including raw sequencing data in the form of BAM files and processed data in the form of Seurat RDS

objects, are accessible through the GEO series accession code GSE277313. All other data that support the findings of this study are listed in Supplementary Data. Source data are provided with this paper.

## Code availability

No custom computational packages or extensive computer code were developed in this study. However, R scripts used to utilize existing packages for this study are available at https://github.com/nick-c-green/Caudate-scMultiome[89].

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

## Acknowledgements

The Collaborative Study on the Genetics of Alcoholism (COGA), Principal Investigators B. Porjesz, V. Hesselbrock, T. Foroud; Scientific Director, A. Agrawal; Translational Director, D. Dick, includes ten different centers: University of Connecticut (V. Hesselbrock); Indiana University (H.J. Edenberg, T. Foroud, Y. Liu, M.H. Plawecki); University of Iowa Carver College of Medicine (S. Kuperman, J. Kramer); SUNY Downstate Health Sciences University (B. Porjesz, J. Meyers, C. Kamarajan, A. Pandey); Washington University in St. Louis (L. Bierut, J. Rice, K. Bucholz, A. Agrawal); University of California at San Diego (M. Schuckit); Rutgers University (J. Tischfield, D. Dick, R. Hart, J. Salvatore); The Children's Hospital of Philadelphia, University of Pennsylvania (L. Almasy); Icahn School of Medicine at Mount Sinai (A. Goate, P. Slesinger); and Howard University (D. Scott). Other COGA collaborators include: L. Bauer (University of Connecticut); J. Nurnberger Jr., L. Wetherill, X., Xuei, D. Lai, S. O'Connor, (Indiana University); G. Chan (University of Iowa; University of Connecticut); D.B. Chorlian, J. Zhang, P. Barr, S. Kinreich, G. Pandey (SUNY Downstate); N. Mullins (Icahn School of Medicine at Mount Sinai); A. Anokhin, S. Hartz, E. Johnson, V. McCutcheon, S. Saccone (Washington University); J. Moore, F. Aliev, Z. Pang, S. Kuo (Rutgers University); A. Merikangas (The Children's Hospital of Philadelphia and University of Pennsylvania); H. Chin and A. Parsian are the NIAAA Staff Collaborators. The authors continue to be inspired by our memories of Henri Begleiter and Theodore Reich, the founding PI and Co-PI of COGA, and also owe a debt of gratitude to other past organizers of COGA, including Ting-Kai Li, P. Michael Conneally, Raymond Crowe, and Wendy Reich, for their critical contributions. This national collaborative study is supported by NIH Grant U10AA008401 from the National Institute on Alcohol Abuse and Alcoholism (NIAAA) and the National Institute on Drug Abuse (NIDA). Tissues were received from the New South Wales Brain Tissue Resource Centre at the University of Sydney, which is supported by the University of Sydney and by the National Institute of Alcohol Abuse and Alcoholism (R28AA012725). Research reported in this publication was also supported by the National Institutes of Health under Award Numbers R01DA053722 (Y.L., H.J.E.), R01AA031176 (Y.L., D.L.), R01AA023797 (Z.P.P.), and R21DA060503 (Z.D.). The content is solely the responsibility of the authors and does not represent the official views of the National Institutes of Health.

## Author contributions

Y.L. and H.J.E. jointly conceived and oversaw the study; X.C. and P.M. performed experiments, supervised by X.X., Y.W., H.G., Y.L., and H.J.E.; N.G. analyzed data and wrote the manuscript, supervised by Y.L., H.J.E., H.G., and J.L.R.; Q.Y. and Z.D. developed and carried out the gene regulatory network analysis; D.L. performed the genotype imputation analysis; G.J. and H.G. performed initial processing of multiome data; J.S. and G.T.S. provided post-mortem samples; A.M.G., P.A.S., Z.P.P., R.P.H., J.A.T., and A.A. contributed ideas during the evolution of the study. All authors contributed to the interpretation and editing of the manuscript.

## Competing interests

The authors declare no competing interests.

## Additional information

[1]Department of Medical and Molecular Genetics, Indiana University School of Medicine, Indianapolis, IN, USA. [2]Center for Computational Biology and Bioinformatics, Indiana University School of Medicine, Indianapolis, IN, USA. [3]Department of Human Cell Biology and Genetics, SUSTech Homeostatic Medicine Institute, School of Medicine, Southern University of Science and Technology, Shenzhen, Guangdong, China. [4]New South Wales Brain Tissue Resource Centre, Charles Perkins Centre and School of Medical Sciences, Faculty of Medicine and Health, The University of Sydney, Sydney, NSW, Australia. [5]Nash Family Department of Neuroscience, Icahn School of Medicine at Mount Sinai, New York, NY, USA. [6]Department of Genetics & Genomic Sciences, Icahn School of Medicine at Mount Sinai, New York, NY, USA. [7]Human Genetics Institute, Rutgers University, Piscataway, NJ, USA. [8]Department of Neuroscience and Cell Biology and The Child Health Institute of New Jersey, Rutgers Robert Wood Johnson Medical School, New Brunswick, NJ, USA. [9]Department of Cell Biology & Neuroscience, Rutgers University, Piscataway, NJ, USA. [10]Department of Genetics, Rutgers University, Piscataway, NJ, USA. [11]Department of Psychiatry, Washington University in St. Louis School of Medicine, St. Louis, MO, USA. [12]Department of Biochemistry and Molecular Biology, Indiana University School of Medicine, Indianapolis, IN, USA. ✉e-mail: edenberg@iu.edu; yunliu@iu.edu

