## [Transparent Peer Review file · Nature Communications]

Integrated Single-Cell Multiomic Profiling of Caudate Nucleus Suggests Key Mechanisms in Alcohol Use Disorder

Corresponding Author: Dr Yunlong Liu

Version 0:

Reviewer comments:

Reviewer #1

(Remarks to the Author)

The authors have addressed many of the questions raised in the previous version and the manuscript has improved. The current version is suitable for publication in Nature Communications.

Reviewer #4

(Remarks to the Author)

- My major methodological criticism of the first submission was in the eQTL analysis and how it linked to AUD GWAS. Those issues have been addressed in the current version of the manuscript.
- My concerns on quality control have also been documented and addressed.
- I would like to re-iterate that this is an important and novel study. The number of subjects and cells is beyond what has previously been published. The additional scale has allowed the researchers to identify new cell type-specific pathways involved in AUD progression. The caudate is also an important hub in AUD that has been understudied. The pathways that the researchers identify are of interest to the AUD community and the dataset itself could be of interest to the broader neurogenetics community. Publishing this dataset, and its analysis, opens up multiple avenues for further study to distinguish which of the observed pathways are likely to play a causal role in AUD progression.
